# Research on Spheroidization of Tungsten Powder from Three Different Raw Materials

**DOI:** 10.3390/ma15238449

**Published:** 2022-11-27

**Authors:** Xiuqing Zhang, Xuchu Hou, Zhenhua Hao, Pei Wang, Yongchun Shu, Jilin He

**Affiliations:** 1Henan Province Industrial Technology Research Institute of Resources and Materials, Zhengzhou University, Zhengzhou 450001, China; 2School of Materials Science and Engineering, Zhengzhou University, Zhengzhou 450001, China; 3School of Materials Science and Engineering, Henan University of Technology, Zhengzhou 450001, China

**Keywords:** radio frequency inductively coupled plasma, spheroidization, tungsten powder, agglomeration

## Abstract

In this work, three kinds of tungsten powders with different particle sizes were spheroidized by radio-frequency (RF) inductively coupled plasma spheroidization. The spheroidization behavior of these tungsten powders was investigated and compared. The spheroidization effects of irregular tungsten powder improves with the decrease in degree of agglomeration and increases with primary particle size. Spherical tungsten powder from irregular powder with a primary particle size of 19.9 μm and an agglomeration coefficient of 1.59 had the best spheroidization effect; its apparent density, hall flow time, and spheroidization ratio are 9.36 g/cm^3^, 6.28 s/50 g, and 98%, respectively. The results show that irregular feedstock tungsten powder with a smaller primary particle size and higher agglomeration degree has a poor spheroidization effect because it is easily affected by the gas flow and deviates from the high temperature zone. On the contrary, irregular feedstock tungsten powder with larger primary particle sizes and lower agglomeration degrees has better spheroidization effects.

## 1. Introduction

Tungsten is a representative refractory metal with many excellent properties, such as a high melting point, good wear resistance, good corrosion resistance, and high thermal conductivity [1,2,3,4]. Furthermore, tungsten has an extremely low vapor pressure not only at room temperature but also at elevated temperatures. Based on these excellent properties, tungsten and its alloys have been widely used in the aerospace, medical, and nuclear industries [5,6,7]. It is worth mentioning that tungsten is massively used as a plasma-facing component in nuclear fusion reactors, being the material of choice for the divertor and sometimes even for the entire vessel, which has been driving research on tungsten materials in the last decade. However, fabrication of tungsten with complex structures is limited by its room temperature ductility and high brittle-to-ductile transition temperature, which is typically between 200 and 400 °C [8]. Nevertheless, additive manufacturing (AM) recently showed the advantage of rapidly manufacturing complex tungsten structures through a layer-by-layer method [9,10]. As a result, AM of tungsten has received a lot of attention [11].

Currently, most of the feedstock materials for AM of tungsten parts are tungsten powder, which has a great influence on the quality of additive manufactured tungsten products [12]. The feedstock tungsten powder requirement for AM involves flowing and spreading well [13,14]. It has been suggested that the feedstock powder of AM should form a uniform packing layer during spreading [15]. Therefore, spherical tungsten powder with good flowability, high purity, and narrow particle size distribution is considered to be the most suitable feedstock material for AM of tungsten. In addition, spherical particles are assumed as the premise in some reported numerical models, such as theoretical models of contact and impact mechanics [16], analytical descriptions of adhesive interaction [17] and advanced models of particle resuspension [18]. Thus, their validation by experiments also requires the use of high-quality spheroidized powder. To sum up, the preparation of high-performance tungsten powder has drawn increasing attention.

At present, RF inductively coupled plasma spheroidization is recognized as the first choice for fabricating spherical tungsten powder due to its high temperature and high enthalpy [19,20,21,22,23]. The mechanism of RF inductively coupled plasma spheroidization is shown in Figure 1. Firstly, a stable plasma torch is established in a vacuum with an RF source. After that, the raw powder is sent into the plasma torch by a carrier gas. When the raw powder particles pass through the high-temperature area of the plasma torch, they are rapidly heated and melted under the action of radiation, convection, conduction, and other mechanisms. Then, spherical droplets are formed under the surface tension. Finally, the spherical droplets leave the plasma medium and enter the quenching chamber, where they are cooled and solidified into spherical powder particles.

The plasma torch established by an RF source has the advantages of high energy density, large heating intensity, etc. In addition, since electrodes are not used as RF power sources, the spheroidized powder will not be polluted due to electrode evaporation. However, the spheroidization of powder has a low utilization rate of heat produced by the plasma torch, only a fraction of which is absorbed by the powder, and its effective power can be calculated as follows [24]:(1)$$P_{E}=\eta P_{in}$$
(2)$$\eta =\left(1-\frac{Q_{1}+Q_{2}}{P_{in}\cdot t}\right)\times 100\%$$
where *P_E_* is the effective power of the plasma torch (kW), *P_in_* is the total power of the plasma (kW), and ƞ is the powder heating efficiency for the plasma torch. *η* is calculated by the following formula. *Q*_1_ is the heat absorbed by the cooling water in the reaction chamber during 1 min, *Q*_2_ is that in plasma torch. The *ƞ* of RF inductively coupled plasma spheroidization system is about 2.2%. In other words, the power used in powder endothermic melting was only about 2.2% of the plasma power. DC plasma was also used for spheroidization. DC plasma was very stable and cheap, but it had shortcomings such as electrode pollution, high operating pressure, and a low ionization degree There are other alternative methods, such as inert gas atomization and the plasma rotating electrode-comminuting process, applied to the preparation of spherical metal powder now. Although the energy cost of RF plasma spheroidizing is relatively high compared with the above methods, its extremely high temperature still makes it the most effective way for the spheroidizing of refractory metal powders with extremely high melting points, such as tungsten.

Many studies have been conducted to prepare spherical tungsten powder by RF plasma spheroidization. Wang et al. [25] investigated the effect of powder feeding rate on spheroidization, and the results show that the spheroidization ratio of tungsten powder can reach 100% at a lower powder feeding rate. Li et al. [26] obtain spherical tungsten powder with a high spheroidization ratio and narrow particle size distribution by RF plasma spheroidization combined with jet milling, which can obtain monodisperse tungsten powder. It is known from the above literature that the properties of the raw material powder are important factors that affect the preparation of spherical tungsten powder by RF inductively coupled plasma spheroidization. However, studies on the spheroidization of tungsten powder from different raw materials has not yet been reported.

In this study, three kinds of tungsten powders with different particle sizes were spheroidized by RF inductively coupled plasma spheroidization. The properties of tungsten powder spheroidized with the same parameters were compared. Furthermore, effects of agglomeration degree of feedstock tungsten powder on spheroidization behavior were also studied.

## 2. Materials and Methods

### 2.1. Experiment

Commercially available irregular tungsten powder (purity > 99.9%; Xiamen Golden Egret Special Alloy Co., LTD., Fujian, China) prepared by hydrogen reduction with different particle sizes, and was used as feedstock powder. The feedstock tungsten powders were named W6, W8, and W20 according to their primary particle sizes.

The feedstock tungsten powder was fed into a self-designed radio frequency inductively coupled plasma spheroidization system to be spheroidized, as shown in Figure 1. The overall radio frequency inductively coupled plasma spheroidization system used in this experiment has been reported in detail elsewhere [27]. As the most critical part of the whole system, the plasma generator consists of a gas distributor head, a powder feeding gun, an inner quartz tube, an outer quartz tube, and an induction coil, as shown in Figure 2 [28]. Gases from three pipes, all of which are argon, are used in this experiment, including carrier gas, central gas, and sheath gas. The carrier gas carried the raw powder into the plasma torch. The central gas was used for ionization to generate inductive plasma. The sheath gas could protect the outer quartz tube from melting due to the high temperature of the plasma. As is shown in Figure 2, when the gas molecules of central gas pass through the inner quartz tube, the plasma is generated at high temperatures of around 3000–10,000 K.

All tungsten powders (W6, W8, and W20) were spheroidized with the same parameters. The powder spheroidized from W6, W8, and W20 were labelled as WS6, WS8, and WS20, respectively. Carrier gas flow rate and plasma power of the spheroidization process were 8 L/min^−1^ and 60 kW, respectively. Finally, the spheroidized tungsten powders were sieved through meshes of 350 and 500 to be further analyzed.

### 2.2. Characterization

Microstructures of the irregular tungsten powder and spheroidized tungsten powder were recorded by using a field emission scanning electron microscope (FESEM, Quanta 250 FEG, FEI). Spheroidization ratio of the spherical tungsten powder R_s_ is obtained by calculating from SEM images by the following Equation (3) [27]:(3)$$R_{s}=\frac{B}{A}\times 100\%$$
where *A* is the total number of tungsten powder particles counted in several SEM images and *B* is the number of spherical tungsten powder particles counted in those SEM images. Specific surface areas of tungsten powder were recorded by an automatic nitrogen adsorption surface analyzer (JW-DX BET, JWGB SCI&TECH, Beijing, China). Primary particle sizes of tungsten powder and spheroidized powder were calculated from its specific surface area. Laser particle size distribution of tungsten powder and spheroidized powder was recorded by a wet laser particle size analyzer (Winner 2308, Ji’nan Winner Particle Instruments, Jinan, China). Flowability and apparent density of the powder were measured by a Scott volumeter (GB/T 1479.1-2011) and a hall flowmeter (GB/T 1482-2010).

### 2.3. Characterization of the Precursor Powder

SEM images and particle size distribution of feedstock irregular tungsten powder are shown in Figure 3. It is seen that the average laser particle size of W6, W8, and W20 are 34.344 μm, 24.821 μm, and 49.130 μm, respectively. It can also be seen from Figure 3a,c that there is a serious agglomeration phenomenon in W6 and W8. However, there are fewer agglomerated particles of W20, as shown in Figure 3e. In order to describe the agglomeration degree of powder more accurately, the agglomeration coefficient A was introduced. The agglomeration coefficient A can be calculated as follows:(4)$$A=\frac{R_{L}}{R_{F}}$$
(5)$$R_{P}=\frac{6}{\rho \cdot S_{w}}$$
where *A* is the agglomeration coefficient and *R_L_* is the secondary particle size measured by a wet laser particle size analyzer. *R_P_* is the primary particle size calculated by Equation (5), where *S_w_* is the mass specific surface area and *ρ* is the density of tungsten. A larger value of the agglomeration coefficient A indicates a higher agglomeration degree for the powder. Table 1 shows the average laser particle size, primary particle size, specific surface area, and the calculated agglomeration coefficients of W6, W8, and W20. It can be seen that the primary particle size of the irregular tungsten powder is 6.28 μm, 7.95 μm, and 19.9 μm, respectively. Agglomeration coefficients of W6, W8, and W20 are calculated to be 5.47, 3.12, and 2.47, respectively. It is indicated that W6 and W8 have a higher agglomeration degree than W20. Table 2 shows the hall flow time and apparent density of the feedstock tungsten powder. The hall flow times of W6 and W8 cannot be measured due to their poor flowability.

## 3. Results and Discussion

Table 2 and Table 3 show the apparent density and hall flow time of W6, W8, and W20 before and after spheroidization. It is obvious that the performance of all tungsten powders is improved after spheroidization. Furthermore, it is seen that the performance improvement of W20 is the most significant, whose apparent density and Hall flow time are 9.36 g/cm^3^ and 6.28 s/50 g, respectively. However, it is generally acknowledged that small particles are easier to melt and spheroidize during plasma spheroidization. Therefore, it can be deduced that a higher agglomeration degree would lead to poor spheroidization. Therefore, W20 with a lower agglomeration degree has a better spheroidization effect than W6 and W8 with severe agglomeration.

A SEM image of the WS6 in different particle size ranges is displayed in Figure 4. It can be observed from Figure 4a that only a small number of particles in WS6 (<30 μm) have been transformed into spherical particles after spheroidization. Theoretically, small particles are more likely to be completely melted and spheroidized. Therefore, the sphericity of spherical particles in WS6 (<30 μm) is high, as shown in Figure 4b, which mainly benefits from its small particle size. However, the spheroidization ratio of WS6 (<30 μm) is only 46.43%. Hence, it can be inferred that agglomerated particles are easily affected by the gas flow and escape from the high temperature zone in the early stage of spheroidization, which makes it difficult to be spheroidized. Particle flow behavior in the plasma torch was mainly affected by inertial force and vicious drag, and the particle Reynolds number is defined as the ratio of the inertial forces to the viscous forces. The relationship between the drag coefficients of non-spherical particles and the sphericity of a particle is listed as follows [28]:(6)$$DC=\frac{24}{Re}\left[1+8.1716exp\left(-0.40655Ø\right)\right]Re^{0.0964+0.5565Ø}+\frac{73.69Re\;exp\left(-5.748Ø\right)}{Re+5.378exp\left(6.2122Ø\right)}$$
where *DC* is the drag coefficient and Ø is the sphericity of a powder. The shapes of agglomerated particles are irregular, the resistance coefficient is greater than that of single particles because of their large contact area with plasma [29], and therefore the agglomerated particles are subject to greater vicious drag. Especially for agglomerated particles with a larger aspect ratio, viscous resistance plays a dominant role in the axial direction [30], thus causing agglomerated particles to be forced to escape from the high temperature area.

The number of spherical particles in WS6 (30–45 μm) is even smaller, as shown in Figure 4c. For WS6 (>45 μm), only a few spherical particles can be observed, as shown in Figure 4d. The spheroidization ratios of WS6 (30–45 μm) and (>45 μm) are calculated to be 33.33% and 28.41%, respectively. It can be inferred that the agglomeration of larger particles in W6 should be taken more seriously. Therefore, this also indicates that a higher agglomeration degree would lead to poor spheroidization.

Figure 5 shows the SEM images of WS8 in different particle size ranges. It can be seen from Figure 5a that WS8 (<30 μm) is composed of a large number of spherical particles and a trace amount of incomplete spheroidization particles. The number of spherical particles decreases in WS8 (30–45 μm) as shown in Figure 5b. Moreover, some spheroidized powder particles with low sphericity can be observed in WS8 (30–45 μm). WS8 (>45 μm) has the fewest spherical particles compared with WS8 (<30 μm) and (30–45 μm), which is shown are Figure 5c. The spheroidization ratio of WS8 (<30 μm), WS8 (30–45 μm), and WS8 (>45 μm) are calculated to be 57.66%, 43.45%, and 30.62%, respectively. Similarly, the spheroidization ratio decreases with increasing particle size. It is seen that WS8 has a higher spheroidization ratio than WS6 in all particle size ranges. Because the agglomeration degree of particles in W8 is lower than that in W6 with the same particle size, the particles in W8 are more regular. Therefore, gas flow has less influence on particles in W8, resulting in the higher spheroidization ratio in WS8. Figure 5d–f shows high magnification images of WS8 (>45 μm). Some spheroidized powder with defects on its surface and some incompletely spherical particles can be observed. During the spheroidization process, agglomerated particles in W8 would continuously collide together to form agglomerate particles with large particle sizes. Furthermore, tungsten has a lower spreading speed and a longer spreading time in comparison with other metals like titanium, copper and, aluminum. Thus, agglomerated particles with large particle sizes need a longer time to shrink and form a spherical powder with high sphericity. This makes the surface of the spherical powder in the WS8 (>45 μm) mostly defective. 

Figure 6 shows the SEM images of WS20 in different particle size ranges. The spheroidization ratios of WS20 (<30 μm), (30–45 μm), and (>45 μm) are calculated to be 96.75%, 62.86%, and 46.88%, respectively. It is obvious that the spheroidization ratio of WS20 is significantly higher than that of WS6 and WS8 in all particle size ranges. It can be seen from Table 1 that agglomeration degree of W20 is lower than W6 and W8. Moreover, some particles in W20 are even monodisperse, as shown in Figure 6c. Therefore, the gas flow has little influence on particles in W20 during spheroidization.

It is worth noting that some nanoparticles attached to the surface of the sphere can be observed in Figure 6b. The nanoparticles are formed by the deposition of the evaporated tungsten during cooling [31,32]. The heavier evaporation behavior also indicates that particles in WS20 could absorb more heat because they would go straight across the high-temperature region of the plasma. Although the majority of WS20 (30–40 μm) and (>40 μm) have been spheroidized after RF inductively coupled plasma treatment, there are still some incompletely spherical particles. Most of these incomplete spherical particles are supposed to be melted during the spheroidization process. Even if the residence time in the high temperature zone is too short for these particles to completely melt and spheroidized, they can still be partially melted. It is indicated that when the agglomeration degree of feedstock powder is lower and the particle size is larger, most of the powder particles could melt because they could go straight across the high temperature zone of the plasma.

Average laser particle size, primary particle size, agglomeration coefficient, and specific surface area of the spheroidized tungsten powder are shown in Table 4. It is noticed that the average laser particle size and primary particle size of W6 increase after spheroidization. It is indicated that the collision and coalescence behavior is heavier when the powder has a higher agglomeration degree and smaller particle size, which leads to an increase in the laser particle size of the spheroidized powder. For W8, average laser particle size and primary particle size have little change after spheroidization. It can be deduced that the majority of particles in W8 that can be completely spheroidized are monodispersed particles. On the contrary, average laser particle size and agglomeration degree of W20 are significantly decreased after spheroidization. It is mainly because W20 has a small number of agglomerated particles. Therefore, most of the particles in W20 can fall into the high temperature region and form spherical tungsten powder during the spheroidization process. Moreover, part of the agglomerated particles in W20 can be spheroidized during the spheroidization process. Another issue is that the evaporation behavior of W20 would be heavier during the spheroidization process.

Figure 7 is a schematic diagram of the influence of agglomeration degree on the spheroidization effect. We consider that particles with a higher agglomeration degree always have a larger aspect ratio and a higher degree of irregularity, which makes it easier to be influenced by the gas flow. Therefore, it cannot enter the high-temperature region of the plasma torch to be melted and spheroidized. The agglomeration degree of the three tungsten powders W6, W8, and W20 used in this experiment gradually decreases, and the number of irregular particles also gradually decreases. Thus. the spheroidization ratio of the three tungsten powders gradually increases in all particle size ranges.

## 4. Conclusions

The effects of the agglomeration degree of the tungsten powder on spheroidization effects were investigated in this research. The results show that irregular feedstock tungsten powder with smaller primary particle size and higher agglomeration degree has poor spheroidization effect because it is easier to be affected by the gas flow and deviates from the high temperature zone. On the contrary, irregular feedstock tungsten powder with a larger primary particle size and lower agglomeration degree has a better spheroidization effect. Spherical tungsten powder from irregular powder with a primary particle size of 19.9 μm and an agglomeration coefficient of 1.59 has the best spheroidization effect; its apparent density, hall flow time, and spheroidization ratio are 9.36 g/cm^3^, 6.28 s/50 g, and 98%, respectively. In the future, an ideal raw powder with controlled particle size and a near-spherical shape will be prepared by spray granulation or other methods to improve the spheroidizing effect. In addition, adding nanoparticles into precursor powder to improve the performance of 3D printed parts will also be explored.

## Figures and Tables

**Figure 1 materials-15-08449-f001:**
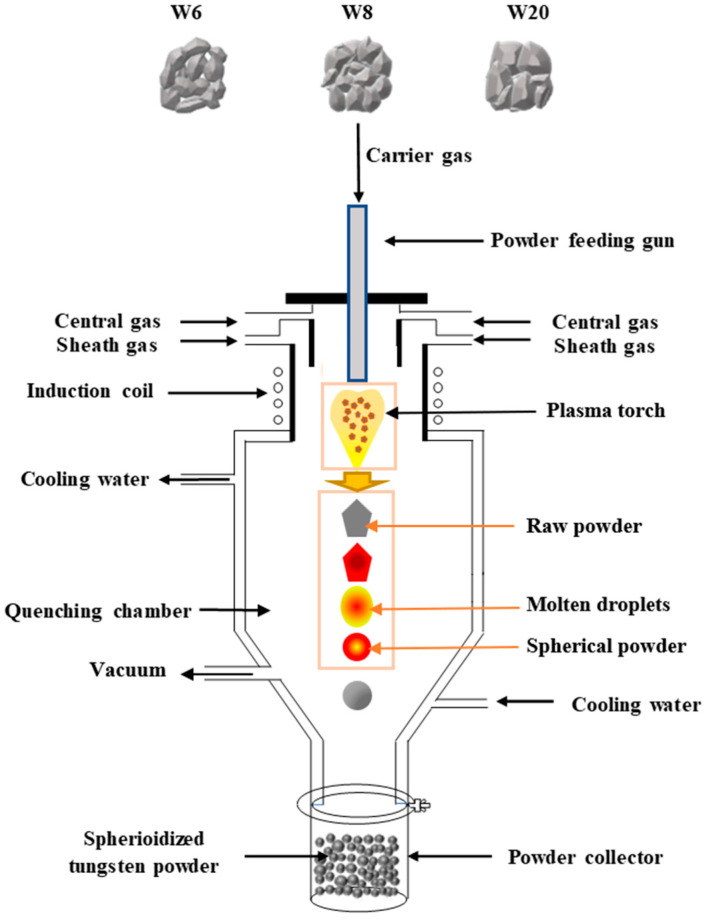
Schematic diagram of the plasma spheroidization system.

**Figure 2 materials-15-08449-f002:**
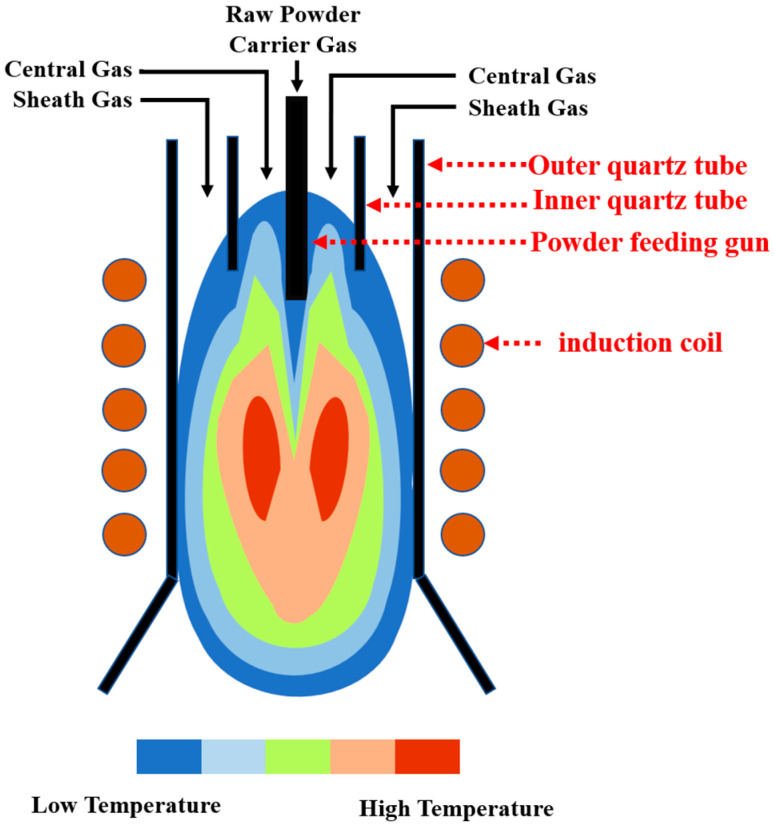
Structure of the RF plasma generator and plasma temperature field of the plasma torch.

**Figure 3 materials-15-08449-f003:**
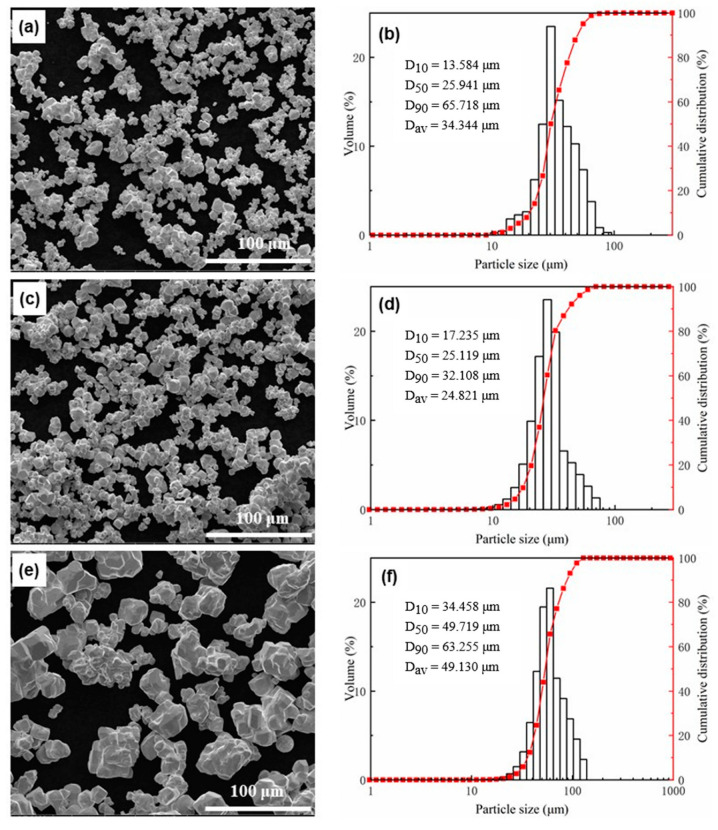
SEM images and particle size distribution of irregular feedstock tungsten powder, respectively, of (**a**,**b**) W6, (**c**,**d**) W8, and (**e**,**f**) W20.

**Figure 4 materials-15-08449-f004:**
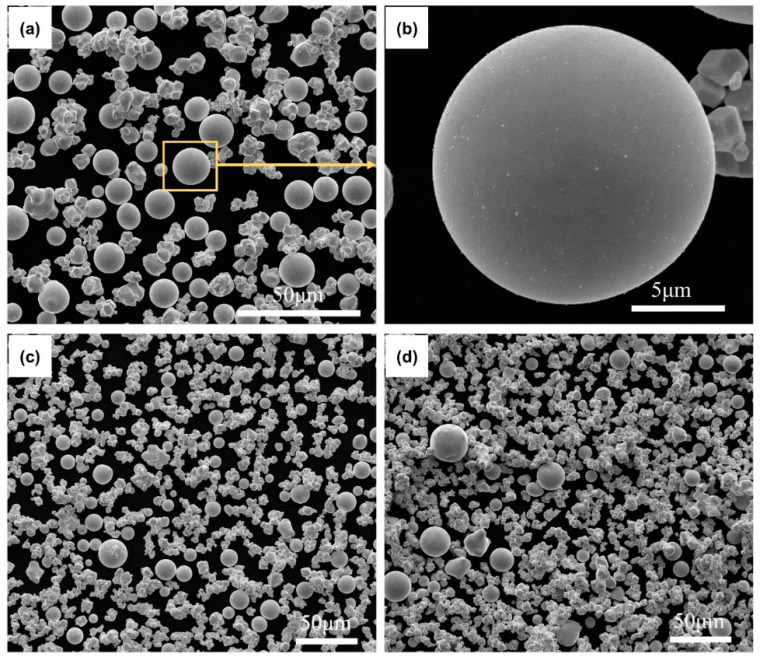
SEM images of WS6 in different particle size ranges: (**a**) (<30 μm), (**b**) high magnification image of the particle in the yellow square in (**a**), (**c**) (30–45 μm), and (**d**) (>45 μm).

**Figure 5 materials-15-08449-f005:**
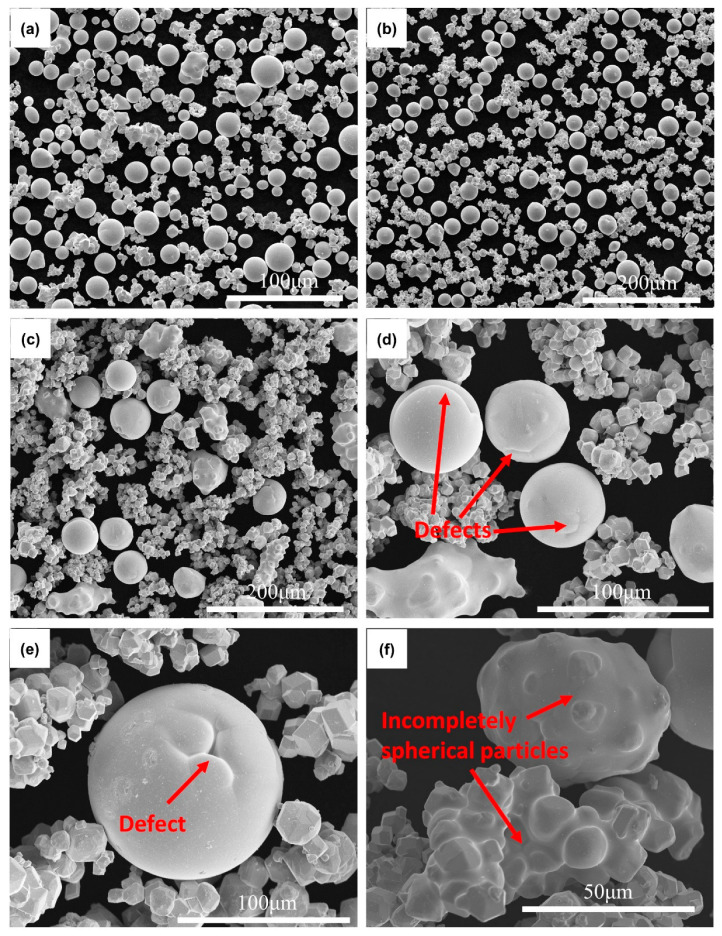
SEM images of WS8 in different particle size ranges: (**a**) (<30 μm), (**b**) (30–45 μm), and (**c**–**f**) (>45 μm).

**Figure 6 materials-15-08449-f006:**
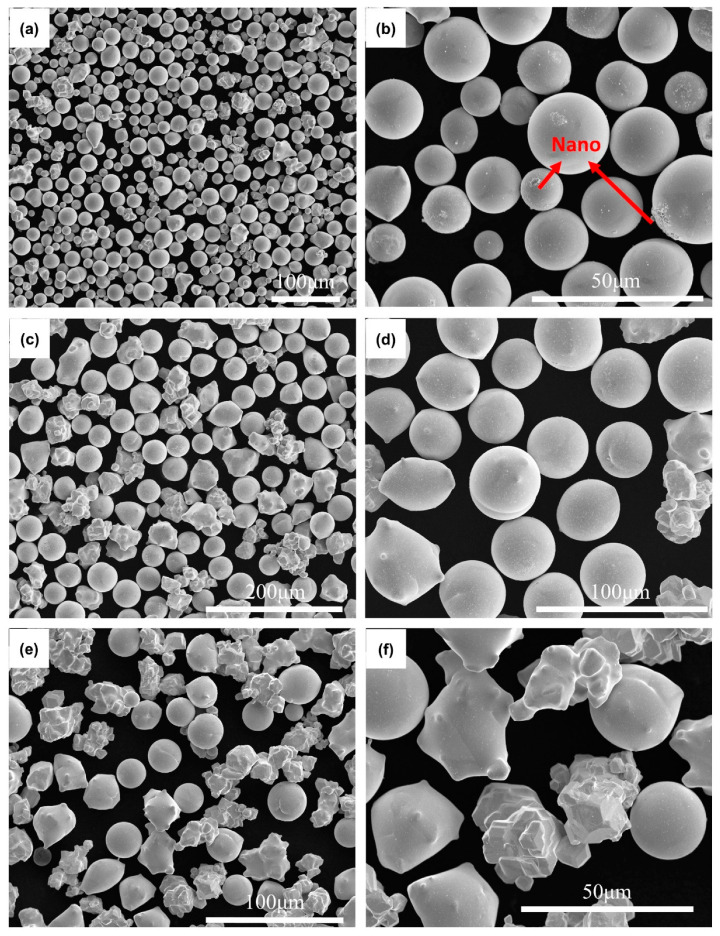
SEM images of WS20 in different particle size ranges: (**a**,**b**) (<30 μm), (**c**,**d**) (30–45 μm), and (**e**,**f**) (>45 μm).

**Figure 7 materials-15-08449-f007:**
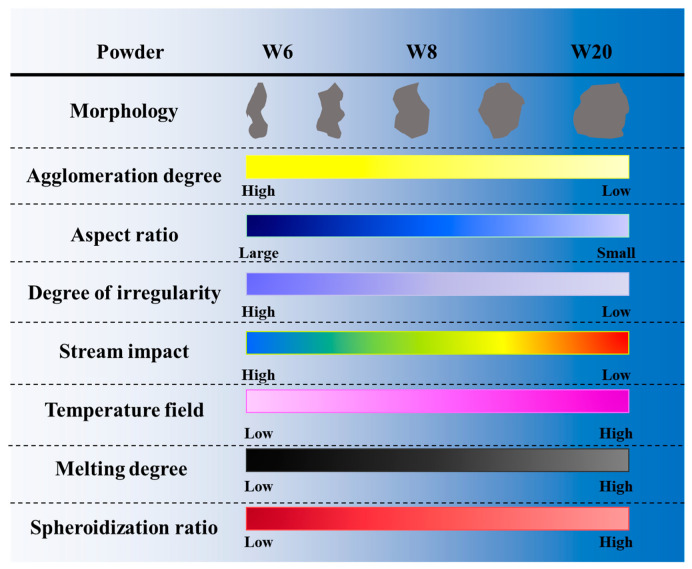
Schematic diagram of the influence of agglomeration degree on the spheroidization effect.

**Table 1 materials-15-08449-t001:** Particle size, agglomeration coefficient, and specific surface area of irregular feedstock tungsten powder.

Powder	Specific Surface Area (m^2^/g)	Primary Particle Size (μm)	Average Laser Particle Size (μm)	Agglomeration Coefficient
W6	0.049	6.28	34.344	5.47
W8	0.039	7.95	24.821	3.12
W20	0.016	19.9	49.130	2.47

**Table 2 materials-15-08449-t002:** Hall flow time and apparent density of feedstock tungsten powder.

Powder	Apparent Density/g·cm^−3^	Hall Flow Time/s·(50 g)^−1^
W6	3.91	-
W8	4.64	-
W20	6.11	11.21

**Table 3 materials-15-08449-t003:** Hall flow time and apparent density of spheroidized tungsten powder.

Powder	Apparent Density/g·cm^−3^	Hall Flow Time/s·(50 g)^−1^
WS6	4.84	19.52
WS8	6.22	16.15
WS20	9.36	6.28

**Table 4 materials-15-08449-t004:** Agglomeration coefficient, primary particle size, laser particle size, and specific surface area of irregular tungsten powder after spheroidization.

Powder	Specific Surface Area m^2^/g	Primary Particle Size/μm	Average Laser Particle Size/μm	Agglomeration Coefficient
WS6	0.029	10.871	36.092	3.32
WS8	0.040	7.769	24.091	3.10
WS20	0.019	16.396	26.105	1.59

## Data Availability

The data used to support the findings of this study are available from the corresponding author upon request.

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
