# Peer review of "Research on Spheroidization of Tungsten Powder from Three Different Raw Materials"

_materials, 2022, doi:10.3390/ma15238449_

Round 1

Reviewer 1 Report

In this work,  the authors have used radio-frequency (RF) inductively coupled plasma sources for spheroidization of tungsten particles. They implemented this plasma technique for three different kinds of raw materials and discussed the degree of spheroidization. Authors should incorporate the following comments to improve the quality of work. 

1. Authors should discuss the mechanism of spheroidization in presence of a plasma medium. It would be better if the authors use some schematic diagrams to understand the complete processes.

2. There should be mention of different gases used in this work. The role of individual gas in the spheroidization process should be discussed in detail.

3. Role of gas flow rate and input power in the  spheroidization process should be discussed in detail. 

4. Authors are using a very high-power rf source in the present study. There should be a calculation for power consumption and mention of merits or demerits of rf source in  spheroidization processes of tungsten. What are alternative ways for the same process?

5. Authors should discuss the results by providing some physical pictures or reactions. There should be a clear picture for the readers to understand the results.

Reviewer 2 Report

please see the attached pdf
